# Quality Evaluation of Bergamot Juice Produced in Different Areas of Calabria Region

**DOI:** 10.3390/foods13132080

**Published:** 2024-07-01

**Authors:** Antonio Gattuso, Rocco Mafrica, Serafino Cannavò, Davide Mafrica, Alessandra De Bruno, Marco Poiana

**Affiliations:** 1Department of AGRARIA, University Mediterranea of Reggio Calabria, 89124 Reggio Calabria, Italy; antonio.gattuso@unirc.it (A.G.); rocco.mafrica@unirc.it (R.M.); serafino.cannavo@unirc.it (S.C.); davidemafrica68@gmail.com (D.M.); mpoiana@unirc.it (M.P.); 2Experimental Station for the Industry of the Essential Oils and Citrus Products SSEA, 89127 Reggio Calabria, Italy; 3Department of Human Sciences and Promotion of the Quality of Life, San Raffaele University, 00166 Rome, Italy

**Keywords:** bergamot, citrus fruit, Fantastico cultivar, flavonoids, nutraceutical potential, polyphenols

## Abstract

Citrus fruits are extensively cultivated worldwide, with Italy and Spain being major producers. In Southern Italy, particularly in Reggio Calabria, a typical citrus fruit is produced, namely, bergamot (*Citrus bergamia Risso* et Poiteau), known for its mysterious origins and exceptional quality essential oil protected by the EU’s PDO (Protected Designation of Origin) designation. Despite historical challenges, bergamot has regained prominence for its nutraceutical potential, especially its flavonoid-rich juice, offering significant health benefits. However, little attention has been paid to understanding the qualitative and quantitative differences of bergamot juice in Calabrian production areas. For this reason, this work aims to investigate the quality characteristics of bergamot juice produced in different areas of Calabria sites. The results showed the best quality attributes of bergamot fruits harvested in the PDO area. In particular, higher levels of total soluble solids, stable acidity, and higher juice were found. In addition, higher contents of ascorbic and citric acids, which are nutritionally valuable and tasteful, were found. The phenolic profile, characterized by the key compounds of bergamot, highlighted the better nutraceutical potential of the fruit grown in the PDO area.

## 1. Introduction

Citrus fruits are among the most widely cultivated fruit crops globally. In Europe (EU), the latest available data from 2022 shows a total production of 10,325 tons, with production strongly concentrated in Mediterranean countries. Italy, second only to Spain, produced 3061 tons [1]. Southern Italy, particularly Calabria and Sicily, is well known for its citrus fruit production. These areas are celebrated for their unique varieties, including blood oranges, lemons, mandarins, and bergamots [2,3].

Bergamot (*Citrus bergamia* Risso et Poiteau), cultivated in the province of Reggio Calabria, is a citrus fruit shrouded in mystery regarding its origins, adding to its enigmatic allure. Some researchers suggest that its origin is derived from bitter orange (*Citrus aurantium*) and lemon (*Citrus limon*) [4], others from bitter orange and lime ((*Citrus aurantifolia* (Christm.)) [5] or a hybrid of citron (*Citrus medica*) and bitter orange [6,7,8,9]. During the last century, this crop experienced a series of events that led to a collapse in the price of bergamot, significantly reducing the presence of this citrus fruit [10,11]. However, today, it enjoys widespread cultivation and popularity. Its success is due to the exceptional quality of its essential oil, produced in a small area of Calabria, obtained by cold-pressing the peel. This quality is unmatched by that obtained elsewhere in the world. In fact, it is protected by the European Union with the PDO quality mark. The compositional differences of the essential oil of bergamot (BEO) produced in the area and outside of this area have been extensively studied. Several researchers, as reported by Dugo et al. [12], have highlighted the differences among the different production areas.

It is widely acknowledged that bergamot essential oil produced elsewhere is not comparable to that of the designated area [13]. The superior quality of bergamot essential oil in the area of Reggio Calabria is attributed to the highly suitable microclimate and soil properties [14,15], which determine the qualitative characteristics of BEO [13].

Until a few years ago, bergamot was only harvested for the production of essential oil, so everything else (juice, pulp, and seeds) was considered industrial waste [16,17]. However, in recent years, studies have revealed its intriguing nutraceutical potential and health benefits for human consumption. Consequently, fresh consumption and the use of bergamot juice have gained significant interest. Compared to other fruits belonging to the same genus, bergamot is abundant in many phytochemicals and nutraceuticals, such as organic acids, limonoids, phenolic acids, and flavonoids [18,19]. Recently, new flavonoids known as 3-hydroxy-3-methylglutaric acid conjugates of naringin and neohesperidin, called melitidin and brutieridin, were discovered. These molecules with statin-like action are very compelling for their remarkable cholesterol-lowering effect [20,21]. Bergamot juice (BJ) has a flavonoid-rich profile, including naringin, neohesperidin, and neoeriocitrin (glycosylated flavanones), which constitute the primary compounds of the polyphenol fraction. Additionally, it contains C-glucosides, flavone O-glycosides, and flavanone O-glycosides [22,23]. For this reason, the consumption of bergamot as fruit has greatly increased, and its nutritional characteristics, along with the juice’s yield, are becoming very important for evaluation.

Despite the general and distinguishing characteristics of bergamot having been studied mainly with regard to the essence, insufficient attention has been paid to the qualitative and quantitative differences in bergamot juice obtained in different Calabrian production areas. Recently, some studies have appeared, and these reported a characterization of fruit quality harvested on the typical area located on the Ionian coast of Reggio Calabria [24,25].

The aim of this study was to evaluate the qualitative parameters of bergamot fruit (Fantastico cultivar), focusing mainly on the juice. Two different growing areas and two harvesting years were considered. Specifically, the fruits were harvested in the province of Reggio Calabria on the Ionian (Melito di Porto Salvo), within the area designed as typical for bergamot essence production and defined as part of the PDO. Additionally, fruits were harvested in an area on the Tyrrhenian side (Rizziconi), which exhibits completely different characteristics in terms of soil and climate.

## 2. Materials and Methods

### 2.1. Fruits Sampling

Bergamot fruits were harvested from “Fantastico” (F) plants grafted onto sour orange trees in two experimental fields located in Melito di Porto Salvo (Ionian coast (IC), PDO area) and Rizziconi (Reggio Calabria, Tyrrhenian site (TS)).

The site of Melito di Porto Salvo, located on the Ionian coast of the Reggio Calabria district, is characterized by an arid climate with higher average monthly temperatures throughout the year compared to the Tyrrhenian side. It experiences less rainfall than the Rizziconi site, with precipitation concentrated during the autumn–winter season. In contrast, rainfall in Rizziconi is more evenly distributed throughout the year. Harvesting took place in two harvest seasons (2022/23 and 2023/24) in December and January by sampling three replicates (twelve fruits each) for both cultivars and sites. The fruits were promptly transported to the Food Technology laboratory at the Mediterranea University of Reggio Calabria for analysis on the same day as harvesting.

The harvest time was selected based on the industrial requirements for processing bergamot for the extraction of essence and juice. Generally, processing occurs between November and February, so two dates within this period were chosen for the study. Bergamot samples were hand-squeezed with a commercial juicer, and the obtained juice was immediately analyzed.

Samples were named “F-M” (Fantastico bergamot fruit harvested in Melito) and “F-R” (Fantastico bergamot fruit harvested in Rizziconi).

### 2.2. Soil Characteristics and Weather Conditions of the Growing Areas

The two growing areas exhibited distinct climatic differences (Figure 1). In Melito P.S., the average annual air temperature was approximately 1.6 °C higher than in Rizziconi. The largest thermal differences between the two areas were recorded during the summer (+1.7 °C) and especially in the autumn months (+2.3 °C). Smaller temperature differences were recorded in the winter (+1.2 °C) and spring (+0.9 °C) months. Significant differences between the two sites were also observed in terms of rainfall. In Rizziconi, the average annual rainfall was more than double that of Melito P.S., with values of 686 mm and 308 mm, respectively. However, it should be emphasized that the difference in rainfall did not affect the vegetative and productive behavior of the bergamot trees, as bergamot, like other citrus fruits, is an irrigated crop. The soil characteristics of the two bergamot groves were very similar. In both sites, the soil texture was sandy clay loam, and the pH was around 6.5–7. The agronomic management of the bergamot trees was identical at both locations.

### 2.3. Chemical Characterization of Bergamot Juice (BJ)

#### 2.3.1. Chemical Determinations

Bergamot Juice was manually extracted using a commercial juicer (Metro Professional GJU2001, METRO Markets GmbH, Düsseldorf, Germany). The juice was analyzed for total soluble solids (TSS) expressed in °Brix using a digital refractometer (DBR 047 SALT, Giorgio Bormac S.r.l., Modena, Italy); pH was determined with a pH meter (Crison Basic 20, Crison Intrument, Barcellona, Spain); titratable acidity (TA) was acquired by titrating juice with 0.25 N NaOH until pH of 8.1, following the International Federation of Fruit Juice Producers (IFU) method [26]. Juice yield (JY) was calculated by dividing the juice weight by the fruit weight and then multiplying it by 100 (JY %).

#### 2.3.2. Ascorbic and Citric Acid Determination

Analysis of organic acids (ascorbic and citric acid) in BJ samples was conducted according to Boninsegna et al.’s method [27]. Briefly, BJ was centrifuged in a refrigerated centrifuge (NF 1200 R, Nüve, Ankara, Turkey) at 10.085 g for 8 min (4 °C), filtered with RC 0.45 µm syringe filter, and diluted with HPLC-grade water (dilution 1:5). The analysis of organic acids was carried out in a HPLC-DAD system (Knauer HPLC Smartline Pump 1000; Knauer Smartline UV Detector 2600, KNAUER, Berlin, Germany) using a SYNERGI HYDRO-RP column (250 mm × 4.6 mm i.d., 4 μm) thermostatically controlled at 22 °C injecting 20 μL of sample. Setting conditions were isocratic elution with a mobile phase solution of potassium phosphate 20 mM acidified (pH 2.9) and a flow rate of 0.7 mL min^−1^. Ascorbic acid was detected at 254 nm and citric acid at 210 nm, and the results were reported as mg of acid per L^−1^ of BJ.

#### 2.3.3. Individual Polyphenols Determination

The quali-quantitative analysis of polyphenols in BJ was conducted using an ultra-high performance liquid chromatography system (UHPLC), as reported by De Bruno et al. [28]. The UPLC PLATINblue system (Knauer, Berlin, Germany) was equipped with a binary pump, Knauer blue orchid column C18 (1.8 µm, 100 × 2 mm) thermostat at 30 °C, and a Photo Diode Array Detector (PDA–1) PLATINblue. A total of 2 µL of filtered BJ sample was injected into the system, and the flow rate was set at 0.4 mL min^−1^. The elution rate was performed using acidified water (formic acid—pH 3.10) in pump A and acetonitrile in pump B with the following gradient: (1) 95% A (0–3 min); 95–60% A (3–15 min); and 60–0% A (15–15.5). The quantification of phenol compounds was achieved using external standards, and the results were expressed as mg 100 mL^−1^ of BJ.

### 2.4. Statistical Analyses

The mean and standard deviation of five measurements were calculated, and analysis of variance (one-way ANOVA) was carried out applying the Tukey post hoc test at *p* < 0.05 by SPSS Software (Version 15.0, SPSS Inc., Chicago, IL, USA).

## 3. Results and Discussion

Although several juice quality parameters, such as carotenoids, were analyzed in a preliminary investigation, the present work reports on total soluble solids, pH, titratable acidity, juice yield, ascorbic and citric acid, and the phenolic profile of the main compounds. The results focused especially on the latter, as bergamot juice has only recently been introduced into the food industry for beverage production due to studies that have demonstrated the important beneficial effects associated with the bioactive compounds.

### 3.1. Qualitative Characteristics of Bergamot Fruits

The sugar content (TSS), pH, and acidity are crucial aspects that define the ripening characteristics of bergamot fruits.

Total soluble solids (TSS) primarily consist of sugars (glucose, fructose, and sucrose) and serve as an indicator of fruit internal quality [29]. TSS content showed statistical differences between the two samples of BJ with higher values in both monitoring times in F-M (Table 1). The same trend was observed in both years considered. Moreover, it is worth noting that in F-M, the TSS remained constant while tending to decrease in F-R during ripening. The annual data showed higher values in F-M during the second year, reaching 11.28 °Brix in December. Data measured on the juices agreed with those reported by Cautela et al. [30]. The TSS observed in fruits from the Ionian area were higher than that reported by Gullo et al. for the same location [24] but were comparable to others reported by the same authors in the same area. The fruits collected in the Tyrrhenian site gave a similar TTS value to that of juice obtained in the same area by Gullo et al. [24]. Differences could be due to the different seasons, as it is possible to observe in this work that the two years of observations release different results.

As reported in Table 1, pH values ranged between 2.35 and 2.62. During the ripening process, no statistical differences in F-M and F-R samples were found during the first year, but in the second year, a different trend was observed, showing a statistical increment in F-M and a decrease in F-R.

TA showed high statistical differences in all ripening stages and in the two considered years, except for F-M in the first year, where, despite a low decrease, no statistical differences were found. In general, the trend followed by acidity is decreasing, as reported by Di Matteo et al. [31], who found the same pattern in lemon fruits. Also, Giuffrè [25] observed the same trend and similar contents for juice obtained from bergamot fruits harvested in the Ionian area. Otherwise, Gullo et al. [24] did not observe pronounced differences between the fruits of the sample collected on the Tyrrhenian site with respect to those from the Ionian, which were in a greater number and with a wide range of variability. With respect to industrially produced juice, the values observed in this work agree with those of Cautela [30].

Generally, juice yield in citrus fruits is considered one of the most important parameters for fruit harvesting and processing. For this reason, various techniques have been developed to enhance juice content [32,33].

The results reported in Figure 2 suggested that during ripening, both samples showed an overall l improvement in juice yield, although with varying statistical differences. In F-M, fruits exhibited a higher JY compared to F-R samples, except in the second-year sampling. This difference could be attributed to the extremely hot and dry weather conditions experienced in the production area during the summer and autumn of 2023. In F-M samples, JY consistently exceeded 40%, reaching 54.51% in the January sampling Year 2022. Conversely, in the F-R sample, JY remained consistently below 40% except in the second year’s sampling, where, despite being relatively high for citrus fruit, it was lower than that found in F-M. Nevertheless, these findings suggest that delayed harvesting may optimize juice extraction and produce riper fruit.

The yield observed in the Ionian site was higher compared to that reported by Giuffrè [25] and Gullo et al. [24]. This could be attributed to various factors, such as different seasons.

### 3.2. Qualitative Characteristics of Bergamot Fruits

The main organic acids detected in BJ through liquid chromatographic analysis were reported in Table 2. They are strongly associated with metabolic plant pathways and are responsible for flavor [34,35]. Ascorbic acid (AA), or vitamin C, is among the vitamins of fundamental importance for humans, as our bodies are unable to synthesize them on their own. AA is widely recognized for its numerous biological functions [36,37].

During monitoring time, AA showed lower values in the F-R sample. The highest concentration was detected in F-M in January 2023 (0.61 g L^−1^). In this year, no statistical differences were observed over time between the monitoring periods. Conversely, in the second year, the AA content decreased during the second monitoring time, with high statistical differences due to organic acid degradation during ripening.

The results of AA are in accordance with Cuzzocrea [38] and with other authors who found similar values in orange varieties [39,40]. The juice obtained from the Ionian area by Giuffrè [25] showed a different trend with a stable or slight decrease between December and January–February, with similar values.

Citric acid (CA) is the primary organic acid in BJ, accumulating within citrus fruit juice sac cells and contributing significantly to the taste and quality of fruits. It is an organic triprotic acid found in abundance in citrus fruits [41]. CA content in BJ varied significantly between the two areas, ranging from 36.87 g L^−1^ in F-R to 49.13 g L^−1^ in F-M.

The comparison between the two juices revealed that the citric acid content during the two different periods of the year remained relatively stable in both samples. The main differences were noted between F-M and F-R in which, as with AA, the highest concentration was detected in F-M.

These differences in organic acid concentrations could be attributable to the temperature effect, which impacts fruit acidity by altering the metabolism and storage of organic acids within vacuoles. Lin et al., 2016 [34] observed that the primary factors affecting open-field fruit quality in winter are cold temperatures and physiological drought.

Low acid concentration in fruits may be linked to higher levels of secondary metabolites such as alcohols and aldehydes, which may adversely affect the flavor of the fruit pre- and post-harvest [42]. The content and concentration variations of organic acids influence taste and play a role in controlling the ripening and storage quality of fruit. The inverse relationship between organic acid levels and fruit weight loss during post-harvest storage emphasizes the importance of organic acids in preserving fruit quality and storability. Their presence can mitigate post-harvest losses, thereby reducing significant economic damage to the citrus industry [43].

These physicochemical findings from bergamot fruits grown in various locations in Calabria highlighted the effect reported in the literature on other citrus fruits. Quality characteristics such as color, acidity, size, juice, and total soluble solids content are strongly influenced by environmental factors such as radiation, relative humidity, and temperature [44].

### 3.3. Comparison of Phenolic Compounds in Bergamot Samples

Bergamot juice has gained increased societal attention in recent years for its nutraceutical and medical effects in preventing and treating various diseases. This is attributed to its high levels of polyphenols [45].

The main phenols detected in BJ (Table 3) showed a chromatographic profile similar to that of bergamot pomace [46]. *p*-coumaric acid and ferulic acid (hydroxycinnamic acids) were recorded in great quantity with values ranging from 1.42 mg 100 mL^−1^ in F-M to 0.61 mg 100 mL^−1^ in F-R for *p*-coumaric acid while ferulic acid from 1.29 mg 100 mL^−1^ in F-M to 0.61 mg 100 mL^−1^ in F-R. Concentrations of *p*-coumaric acid followed different trends in both samples for both years, exhibiting an ever higher content in BJ F-M.

The higher content of ferulic acid was also detected in F-M between the analyzed samples in the two harvest years. Values were stable (no statistical differences) between the first and the second monitoring time, except for F-R juice, which, during the second year, observed a statistical (*p* < 0.01) reduction.

The other compounds detected belong to the flavonoid. Small quantities of eriocitrin and narirutin were found, and both flavonones showed similar behavior with a statistical increase in F-M in the first year only, confirming a stable level (no statistical differences) in F-M in the second year and in F-R in both years.

Neoeriocitrin, naringin, and neohesperidin were detected in high concentrations in BJ with values comparable to those reported by Da Pozzo et al., 2018 [47].

Neoeriocitrin was lower in F-R only in the first monitoring time of the second year (9.87 mg 100 mL^−1^ in F-M, 8.59 mg 100 mL^−1^ in F-R). For the other two major flavonoids, naringin and neohesperidin, the differences were considerable between the sample juices, reaching up to twice the F-R content in F-M. The concentration levels of naringin (values comprised between 6.45 and 13.88 mg 100 mL^−1^) showed a statistical increment in F-M in the first year and stable levels in the second year. BJ obtained in Rizziconi did not exhibit changes from the first to the second measurement. Similar trends were observed for neohesperidin without any variation between different times and years. Statistical analysis showed relevant differences (*p* < 0.01) between F-M and F-R, where the content is almost always reduced by half. For instance, in January 2024, 12.11 mg 100 mL^−1^ were found in F-M and 4.07 mg 100 mL^−1^ in F-R. 

Melitidin and brutieridin, C-Glycosyl flavones, are characteristic of bergamot fruits and well-known statins for their anticolesterolemic activity [48]. The statistical differences showed a clearly higher concentration in the F-M sample than in F-R in both years and in both harvesting times. In particular, brutieridin was at least twice the concentration in F-M, and during the second half of the second year, it tripled.

The compositional differences in BJ samples obtained from the fruits of the two different areas are highlighted by the total polyphenols content of the main quantified phenols, which showed that in all periods in which the BJ was analyzed, the polyphenol content was significantly (*p* < 0.01) lower in F-R, as clearly visible in Figure 3.

## 4. Conclusions

The results of this two-year study, conducted during two different harvest times (December and January), demonstrate how the production area, highly suited to the cultivation of bergamot, strongly influences the characteristics of the fruit. Although it was already known that the characteristics of the essence find their maximum qualitative expression in the designated Ionian coastal strip of the province of Reggio Calabria, it has been shown that the physicochemical characteristics of the fruit are also strongly influenced by the area, and thus by the microclimatic conditions. Juice quality is influenced by these parameters, particularly temperature. In fact, the temperature appears to be higher at the Melito Porto Salvo site (about 1.6 °C) than at the Rizziconi site. As for rainfall, on the other hand, it has a limited influence because in the absence of rainfall, irrigation was used. Even if the seasonal variability could be influencing the characteristics, juice yields, the content of organic acids, and flavonoids for which there is growing nutraceutical and health interest showed a higher quality in the Melito Porto Salvo area than in Rizziconi, even though both areas are in the Calabria region (Italy).

## Figures and Tables

**Figure 1 foods-13-02080-f001:**
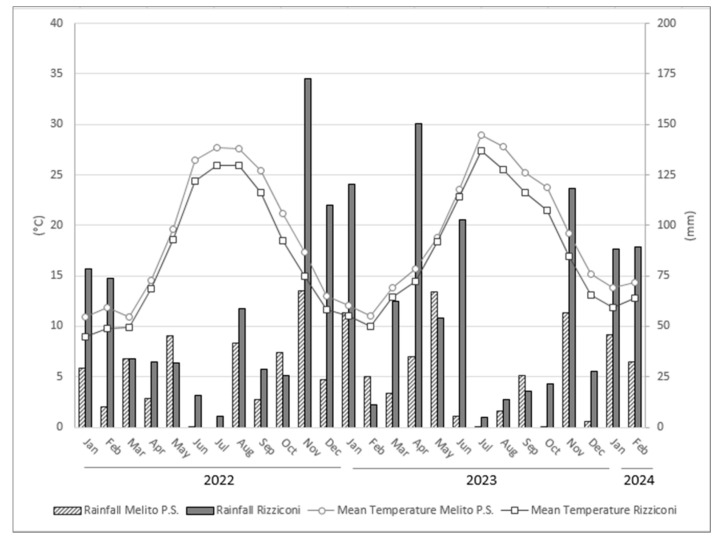
Monthly changes of mean temperatures and rainfall during years 2022, 2023, and 2024 of the two experimental sites.

**Figure 2 foods-13-02080-f002:**
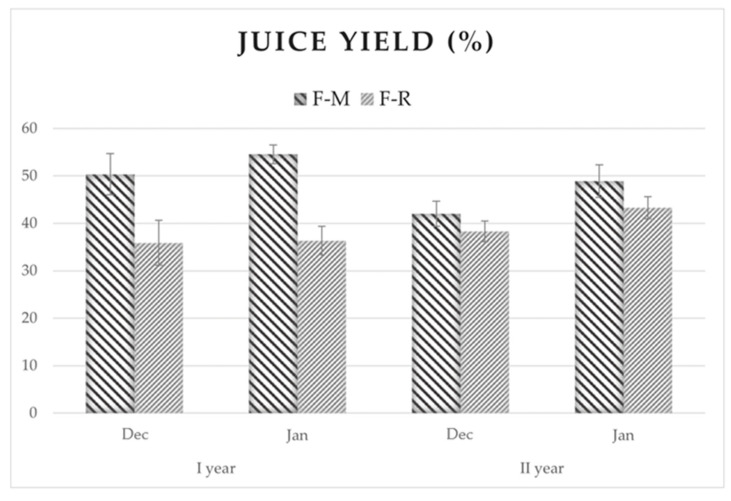
Juice yield in bergamot fruits.

**Figure 3 foods-13-02080-f003:**
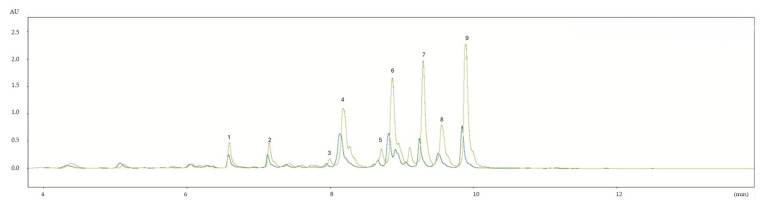
Chromatographic profiles comparison of F-M and F-R BJ samples (UHPLC). (1) p-coumaric acid; (2) ferulic acid; (3) eriocitrin; (4) neoeriocitrin; (5) narirutin; (6) naringin; (7) neohesperidin; (8) melitidin; and (9) brutieridin. Mustard color: F-M sample; green: F-R sample.

**Table 1 foods-13-02080-t001:** TSS, pH, and TA in bergamot samples.

**I YEAR**	**TSS (°Brix)**			**pH**			**TA (g L^−1^ Citric ac. m)**		
	**Dec**	**Jan**	**Sign**	**Dec**	**Jan**	**Sign**	**Dec**	**Jan**	**Sign**
F-M	10.36 ± 0.13	10.24 ± 0.19	n.s.	2.52 ± 0.09	2.57 ± 0.07	n.s.	49.28 ± 1.07	48.58 ± 1.21	n.s.
F-R	8.84 ± 0.54	7.7 ± 0.21	**	2.59 ± 0.06	2.61 ± 0.1	n.s.	43.65 ± 1.78	41.23 ± 1.11	**
Sign	**	**		*	**		**	**	
**II YEAR**	**TSS (°Brix)**			**pH**			**TA (g L^−1^ Citric ac. m)**		
	**Dec**	**Jan**	**Sign**	**Dec**	**Jan**	**Sign**	**Dec**	**Jan**	**Sign**
F-M	11.28 ± 0.13	11.04 ± 0.53	n.s.	2.35 ± 0.02	2.62 ± 0.08	**	55.37 ± 1.86	49.04 ± 0.55	**
F-R	8.74 ± 0.25	7.58 ± 0.18	**	2.57 ± 0.06	2.48 ± 0.02	**	50.09 ± 1.48	42.56 ± 3.06	**
Sign	**	**		**	**		**	**	

Statistical analysis significance based on Tukey’s post hoc test: ** for significance at *p* > 0.01; * for significance at *p* > 0.05; and n.s. for not significant.

**Table 2 foods-13-02080-t002:** Ascorbic (AA) and citric (CA) acids in bergamot juice (g L^−1^).

**I YEAR**	**AA**			**CA**		
	**Dec**	**Jan**	**Sign**	**Dec**	**Jan**	**Sign**
F-M	0.57 ± 0.04	0.61 ± 0.07	n.s.	46.44 ± 2.64	49.13 ± 3.14	n.s.
F-R	0.43 ± 0.07	0.48 ± 0.02	n.s.	38.87 ± 3.24	40.35 ± 2.04	n.s.
Sign	**	**		**	**	
**II YEAR**	**AA**			**CA**		
	**Dec**	**Jan**	**Sign**	**Dec**	**Jan**	**Sign**
F-M	0.59 ± 0.02	0.48 ± 0.06	**	47.96 ± 1.08	47.46 ± 2.2	n.s.
F-R	0.42 ± 0.02	0.32 ± 0.04	**	47.25 ± 2.98	44.8 ± 2.54	n.s.
Sign	**	**		n.s.	n.s.	

Statistical analysis significance based on Tukey’s post hoc test: ** for significance at *p* > 0.01; and n.s. for not significant.

**Table 3 foods-13-02080-t003:** Polyphenols composition of BJ samples.

mg 100 mL^−1^		I YEAR			II YEAR		
		F-M	F-R	Sign	F-M	F-R	Sign
*p*-coumaric acid	Dec	1.02 ± 0.02	0.83 ± 0.03	**	1.42 ± 0.08	1.14 ± 0.08	**
	Jan	1.09 ± 0.01	0.73 ± 0.13	**	1.22 ± 0.1	0.75 ± 0.09	**
Sign		**	n.s.		n.s.	**	
Ferulic acid	Dec	0.98 ± 0.18	0.78 ± 0.09	n.s.	1.29 ± 0.14	0.88 ± 0.06	**
	Jan	0.98 ± 0.14	0.66 ± 0.1	**	1.06 ± 0.13	0.61 ± 0.09	**
Sign		n.s.	n.s.		n.s.	**	
Eriocitrin	Dec	0.57 ± 0.05	0.6 ± 0.15	n.s.	1.27 ± 0.08	0.74 ± 0.04	**
	Jan	0.68 ± 0.06	0.56 ± 0.09	n.s.	1.24 ± 0.02	0.62 ± 0.09	**
Sign		*	n.s.		n.s.	n.s.	
Neoeriocitrin	Dec	9.62 ± 0.83	9.13 ± 0.38	n.s.	9.87 ± 0.34	8.59 ± 0.55	*
	Jan	10.89 ± 0.68	12.15 ± 1.88	n.s.	11.07 ± 1.61	9.37 ± 1.25	n.s.
Sign		n.s.	*		n.s.	n.s.	
Narirutin	Dec	0.22 ± 0.05	0.22 ± 0.05	n.s.	1.21 ± 0.02	0.85 ± 0.12	**
	Jan	0.37 ± 0.01	0.15 ± 0.03	**	1.26 ± 0.45	0.76 ± 0.07	n.s.
Sign		**	n.s.		n.s.	n.s.	
Naringin	Dec	9.69 ± 1.55	6.66 ± 0.85	*	13.44 ± 0.6	7.35 ± 0.26	**
	Jan	12.49 ± 1.25	8.49 ± 1.83	*	13.88 ± 0.99	6.45 ± 1	**
Sign		*	n.s.		n.s.	n.s.	
Neohesperidin	Dec	9.12 ± 2	4.1 ± 1	**	11.93 ± 0.6	5.04 ± 0.41	**
	Jan	8.92 ± 1.2	5.15 ± 1.22	**	12.11 ± 1.92	4.07 ± 0.64	**
Sign		n.s.	n.s.		n.s.	n.s.	
Melitidin	Dec	3.99 ± 0.72	2.05 ± 0.77	**	8.26 ± 0.47	4.67 ± 0.35	**
	Jan	5.09 ± 0.83	2.14 ± 0.39	**	8.55 ± 1.06	3.48 ± 0.25	**
Sign		n.s.	n.s.		n.s.	**	
Brutieridin	Dec	21.74 ± 2.2	9.56 ± 2.85	**	23.05 ± 0.66	10.97 ± 0.72	**
	Jan	24.53 ± 0.69	8.56 ± 1.28	**	22.19 ± 2.38	7.21 ± 0.43	**
Sign		n.s.	n.s.		n.s.	**	
Total Polyphenols	Dec	56.95 ± 6.01	33.93 ± 4.93	**	71.75 ± 2.58	40.23 ± 1.98	**
	Jan	65.04 ± 3.53	38.6 ± 6.5	**	72.59 ± 8.47	33.31 ± 3.05	**
Sign		n.s.	n.s.		n.s.	*	

Statistical analysis significance based on Tukey’s post hoc test: ** for significance at *p* > 0.01; * for significance at *p* > 0.05; and n.s. for not significant.

## Data Availability

The original contributions presented in the study are included in the article, further inquiries can be directed to the corresponding author.

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
