# Peer review of "Quality Evaluation of Bergamot Juice Produced in Different Areas of Calabria Region"

_foods, 2024, doi:10.3390/foods13132080_

Round 1

Reviewer 1 Report

Comments and Suggestions for Authors

Quality evaluation of Bergamot fruits produced in different areas of Calabria Region (Manuscript number foods-3043004) is comparative study of Bergamot fruits from two different areas from Italy at two harvesting seasons for two consecutive years. It is an interesting study to compare the fruit quality in terms of juice yield, and juice qualities for different harvesting seasons and harvesting year. The authors did extensive experimental works to fulfill their aims of this study, while they also conducted enough discussion of the results. However, I am a single major comment – the expression of the authors messages in this manuscript is a bit difficult to understand. Therefore, I suggest to revise its English language from a professional language correction service; the authors may use MDPI English Revision service. Apart from that, following few minor corrections are suggested:

Abstract: please explain the abbreviations DPO and PDO.

Fig. 1: December should be abbreviated as Dec; not Dic!

Table 2: December should be abbreviated as Dec; not Dic! In “p<0.05”, p should be italicized (whole manuscript).

Table 2: December should be abbreviated as Dec; not Dic! In “p<0.05”, p should be italicized (whole manuscript).

Table 3: December should be abbreviated as Dec; not Dic! In “p<0.05”, p should be italicized (whole manuscript).

Consistency in listing references, more specifically, in terms of title, italicized scientific names etc.

English language needs extensive revision to make it clear enough to understand what the authors wanted to express and how the processes were conducted.

Comments on the Quality of English Language

Extensive English language correction is recommended.

Reviewer 2 Report

Comments and Suggestions for Authors

The manuscript foods-3043004 has a potential but it must be improved. It is insufficient to report the differences in the quality of Bergamot fruits, which are expected, depending on the locality where the fruit was sampled, and not to take into account the quality (composition and characteristics) of the soil, the amount of precipitation in the season, the number of sunny days, etc., and correlate it with obtained quality parameters. If the authors have these data, it is necessary to introduce them into the work and do a major revision. In this way, the quality of work would be deepened. If not - the work in this form is not of satisfactory quality for a journal of this rank.

Besides,

·      Account for abbreviations - DPO in the abstract write what the abbreviation is, PDO (line 46) write what the abbreviation is, BJ is introduced in line 63, but appears in the entire text both the full name and abbreviation together (e.g. line 102, 238, …)

·     I suggest deleting the S1 and S2 tables because the same data is already shown in the paper in tables 1-3 and figure 1. What is the purpose of tables S1 and S2 at all? They are not mentioned in the text and their existence duplicates the results.

·     Dicember” should be “December” (line 92) and “Dic” should be “Dec” (table 1,2, and 3)

Comments on the Quality of English Language

 Minor editing of English language required

Reviewer 3 Report

Comments and Suggestions for Authors

The work described in this manuscript is very interesting and relevant as it shows the chemical characteristics of bergamot juice from fruit cultivated in two sites of the DPO area. The work was well designed and performed. The data are good.

 -Overall, the study is partially original because the chemical composition of bergamot juice has been study previously, including bergamot juice from the same area.

-The title should include the word “juice”, as the study focused on juice rather than the fruit. I missed.

-Could you describe how the harvest time was determined in both sites?

-I missed the climatic conditions during the two years, as the climatic conditions impact the chemical composition of fruits.

-I missed other quality attributes of juice, for example, the color and the content of other beneficial compounds like chlorophyll/carotenoid content.

-You should modify the subtitle for section 3.2. Perhaps, “Content of ascorbic and citric acids”.

-Please, review the style and presentation of the manuscript. I found some minor mistakes, for example: It should be “fields” in line 85; It should be “both sites” in line 93; Please, provide the centrifugation speed in “g” not in “rpm”; please, review line 145. Results and discussion section is in some parts confusing/wordy.

Round 2

Reviewer 2 Report

Comments and Suggestions for Authors I thank the authors for the comments accepted and the corrections made. The manuscript has been significantly improved and is ready for publication.

Author Response

The Authors thank the Reviewer